# Staff Perspectives: Defining the Types, Challenges and Lessons Learnt of University Peer Support for Student Mental Health and Wellbeing

**Julia Pointon-Haas** [1,*], **Nicola Byrom** [1] , **Juliet Foster** [1], **Chloe Hayes** [1] **and Jennifer Oates** [2]

1 Department of Psychology, King's College London, London SE1 1UL, UK; nicola.byrom@kcl.ac.uk (N.B.); juliet.foster@kcl.ac.uk (J.F.); chloe.m.hayes@kcl.ac.uk (C.H.)
2 Faculty of Health & Medical Sciences, University of Surrey, Surrey GU2 7YH, UK; j.oates@surrey.ac.uk
* Correspondence: julia.a.pointon-haas@kcl.ac.uk

**Abstract:** In university settings, peer support brings people together based on their student identity. Peer support has been advocated as an innovative intervention to aid student mental health and wellbeing as part of a whole university approach, especially post-pandemic when student support is critical. While the literature describes three types of university peer support for student mental health and wellbeing, peer-led support groups, peer mentoring, and peer learning, the sector lacks agreed definitions for these interventions. Formal reporting on peer support initiatives is rare, suggesting further types of peer support practice may exist. This qualitative study, comprising semi-structured interviews with 16 university staff members at 14 different institutions, aimed to generate comprehensive definitions of the types of peer support used in the sector through template analysis. The study also sought to understand the current practice, experiences, and challenges around implementing peer support interventions for undergraduate and postgraduate students' mental health and wellbeing from the perspective of the staff who support and coordinate these programmes. Five types of peer support were identified and defined. In addition, the challenges of engagement, resource and capacity, and evaluation were highlighted. Finally, lessons learnt provided potential ways to address the challenges outlined and provided sector guidance for further developing peer support as part of a whole university approach to student mental health and wellbeing.

**Keywords:** peer support; university student mental health; pandemic; college students; mental health; whole university approach

## 1. Introduction

Universities need new ways to support students after the pandemic. Students' mental health and wellbeing concerned many before the onset of COVID-19 [1,2], given their susceptibility to poor mental health as a population [3,4]. The pandemic exacerbated mental health issues [5], increasing anxiety levels and reducing subjective wellbeing for UK university students in 2021 compared to 2019 [6]. It was apparent during the pandemic that students were an at-risk population facing added challenges [7]. If adequate efforts are not made to address the mental health needs of university students beyond the pandemic, this population could experience long-term consequences [8]. The mental health implications for students post-pandemic are significant [9]. Innovative approaches to support them are required.

Many universities are considering proactive interventions as part of a whole university approach. The whole university approach, advocated for globally [10–13], is multifaceted, recognising that every aspect of university life can affect all community members' mental health and wellbeing [14,15]. The approach is gaining momentum in the UK with the 'University Mental Health Charter,' which outlines peer support as a proactive intervention

that can improve wellbeing [14,16] as part of an integrated system of student mental health care [17,18].

Peer support may be a valuable intervention for improving young people's mental health, especially university students [2,16,19]. Peer support is 'support provided by and for people with similar conditions, problems or experiences' [20] (p. 2). In the university context, as part of a whole university approach, peer support refers to students supporting each other as 'peers' because of their shared university experience. Students turn to each other informally for support and are far more likely to seek support from peers than professional services [21–24]. If students already turn to each other, formal, structured interventions based on peer support may enhance the quality of the support peers provide each other. Students report finding peer support easy to use and a reliable additional support to professional services [25].

There are multiple types of formal peer support, including peer-led support groups, peer mentoring, and peer learning [26]. Peer-led support groups unite students for mutual support, peer mentoring connects higher-year/more experienced students to lower-year/less experienced students, and peer learning convenes students based on academic objectives [26]. In addition to the structure of these interventions varying dramatically, their objectives are also broad. For example, peer-led support groups are delivered with disparate objectives, such as helping students from marginalised backgrounds transition into university or supporting students with depression [26].

Further complicating the challenge of understanding peer support in the university context, the range of peer support captured within the published literature is not exhaustive. Although some types of peer support are represented in the literature with relevant outcome measures [26], others exist in practice, which have not measured mental health or wellbeing. Known examples of other peer support types without demonstrated impact on mental health or wellbeing are peer education [27], peer counselling [28], and peer health education [29]. Even within the peer support literature, precise definitions, objectives, and evaluations are lacking, hindering comparisons of peer support types [30–33]. Without exhaustive reporting, it is difficult to grasp the considerable nuance of implementing peer support or know if interventions are achieving their desired goal.

In addition to better understanding the impact of specific types of peer support, there is a need to investigate the training [19] and support for peer supporters. Formal peer support, harnessing a natural student drive to support each other, creates structure, allowing for consistency and seeking to ensure safety. The structure of formal peer support requires professional staff engagement. While young people tend to turn to each other for help, they generally lack the skills and knowledge to support and signpost a peer needing help, so training is required from staff [34].

University peer support requires more from professionals than just training, however. Staff also tend to act as commissioners and organisers behind the scenes, championing peer support and securing resource to run it. Peer support is not automatically self-sustaining; students require training and ongoing support from skilled and committed professionals to ensure organisation, continuity, safety, and structure for sustainable delivery [34]. While the staff roles coordinating peer support are diverse, their positions are critical. However, staff involvement in university peer support is largely unreported. Literature outlining peer support interventions for student mental health and wellbeing does not account for the staff perspective [26]. No published literature highlights the experiences of staff coordinating different types of peer support for student mental health and wellbeing. Without this understanding, the sector lacks shared knowledge of how these peer support programmes operate.

Although developing shared knowledge of peer support is helpful, the higher education sector is diverse, so context needs to be accounted for. For example, universities in the UK include five types: ancient universities, red brick or civic universities, plate glass or 1960s universities, new universities (post-1992) or polytechnic and metropolitan universities, and Russell group universities [35]. These categories are not well-defined,

however. Some lists include the University of London, a federal public research university comprising 17 universities, as another category [36]. The varying types of UK universities also have a range of academic and social differences among them. Some examples include entry requirements, workload, emphasis on research, university structure, accommodation, and being city- or campus-based [37]. The diverse range of universities, with their varied cultures, also tend to have different student populations. Students from private schools with fees are seven times more likely to be accepted into Oxford or Cambridge than those in 'non-selective' state schools and more than twice as likely to attend highly ranked Russell Group institutions [38]. Therefore, when seeking to understand university peer support in practice, it is essential to consider the range of UK universities and account for the sector's context through a representative sample.

Thus far, universities have been actively encouraged to develop peer support interventions [14] without any clear consensus [30] or solid evidence [39] around what constitutes effective peer support in their university context. Although some forms of university peer support are feasible, acceptable, and safe, implementation difficulties still exist [40]. The sector needs an in-depth understanding of all types of peer support to implement successful interventions. Without a roadmap, it is challenging for universities to consider how they might incorporate peer support as a proactive intervention that is part of a whole university approach.

This study aimed to understand university peer support for undergraduate and postgraduate students' mental health and wellbeing through the perspective of staff who support and coordinate these programmes. With this, the following research question guided the study: What are the approaches, challenges, and lessons learnt from running peer support for university mental health and wellbeing from the staff perspective? Therefore, this study helped to further define peer support, its objectives, and its various types in practice by exploring the successes and challenges of implementing programmes from the unique view of staff representing a diverse sample of UK universities.

## 2. Materials and Methods

Semi-structured interviews took place from June to August 2021 after gaining ethical approval (MRSP-20/21-23031) to explore staff experiences and insight [38]. Staff who coordinated university peer support for student mental health or wellbeing were invited to participate. Beyond stating that formal peer support approaches, from group interventions to one-to-one structures, were of interest, no further limitations or definitions were set regarding the types of peer support being explored. Student Minds, the UK's student mental health charity, has a significant history of supporting peer support in universities, including the provision of training for staff to train, supervise, and support student peer facilitators. The charity assisted recruitment for this study, sharing information with staff across their network.

Staff were recruited via email and social media. An email was sent through the UK's Student Mental Health Research Network, SMaRteN, the UK Healthy Universities Network, the International Academic Peer Learning Network, the Scottish Peer Support Network JISCmail, Student Minds, and researcher contacts, including those made on LinkedIn. After invitations were sent, those who wished to participate contacted the researcher to arrange an interview date and time.

Non-probability volunteer sampling was chosen so that staff could self-identify as running peer support for student mental health or wellbeing and participate [41]. There is little consensus on the definitions or types of university peer support in the sector [26], making the population that coordinated programmes hard to find. While some staff networks exist, such as the International Academic Peer Learning Network [42] or the Scottish Peer Support Network [43], no organisation brings together any staff coordinating peer support in the UK. Although volunteer sampling risked participants coming forward who could not provide sufficient information [44,45], it enabled access to a population that would have been hard to identify through other sampling methods. Those interviewed

were asked if they knew of others to talk to, so snowball sampling was also incorporated into this study [41,44].

The dataset comprised 15 interviews with 16 participants from 14 universities. Participants included university staff in the UK and staff partners of Students Minds. Participants were assigned a number to delineate different universities. They were assigned additional letters when multiple participants represented the same university. For example, one interview had two participants, 2A and 2B. Two interviews were conducted with participants running different programmes at the same university in different departments: 14A and 14B. All participants except two were non-academic staff: 7 and 10.

Higher education institutions in the dataset represented five types of UK universities outlined in Table 1. Additionally, one university was affiliated with Student Minds as an international university. No civic universities (red brick) were represented [35]. Thus, a relatively representative sample was gathered with an even split in location. Six were campus-based, and eight were city universities.

**Table 1.** Number of UK University Types in Dataset.

| Ancient | Federal Public Research | Plate Glass/1960s | New/Post-92 | Russell Group |
|---------|------------------------|-------------------|-------------|---------------|
| 2 | 1 | 5 | 3 | 2 |

Following informed consent, semi-structured interviews were conducted and recorded on Microsoft Teams, allowing follow-up questions to create rich discussions. The interview topic guide (Supplementary Material S1) was developed from a systematic review of university peer support, highlighting knowledge gaps in definitions, intervention characteristics, and objectives [26]. Participants were asked what they believed peer support to be. The discussion explored participants' understanding of peer support based on their experiences and knowledge.

Analysis was conducted from a 'contextual constructionist' position, recognising that knowledge is local, provisional, and situationally dependent [46,47]. The stance recognises multiple interpretations, which can be made based on the researcher's positionality and the broader social context [47]. In this study, the principal researcher had an established background of 12 years coordinating university peer support programmes. As this epistemological position considers the researcher's context, it was appropriate to apply.

Template analysis was employed, providing a 'middle ground' approach to thematic analysis [48], seeking to understand a set of experiences across a data set [49]. While template and thematic analyses overlap, conceptualising codes, themes, and the analytic process differ [50]. Recordings were manually transcribed (JPH, CH), and transcripts were coded according to the steps outlined by King [48]. Coding and analysis occurred after all the interviews were complete. Following initial familiarisation and analysis of two interviews, a coding template was developed in consultation with JPH and CH. The codes of 'challenges' and 'types of peer support' were used initially according to familiarity with the interviews and the research question. After five interviews, further consultation occurred with the broader research team to agree on a final framework (Supplementary Table S1). The lead researcher applied this template to the remaining ten interviews and modified it as necessary.

Codes could be descriptive or interpretative [51] and were used to develop themes [50]. The deductive development of a coding template was combined with an inductive approach, allowing the in-depth development of specific themes based on the richness of relevant data [52]. For example, the initial coding template included types of peer support expected to be found based on literature [26]. Still, additional codes were added to capture further types of peer support outlined by participants. Emerging themes were organised by relationships, with themes such as 'impact' and 'defining peer support' identified as relating to one another. After all interviews were coded, the template was refined by assessing each code and merging or deleting codes to create one final template of themes.

The quality of the analysis was appraised by checking the study against accepted criteria: credibility, dependability, confirmability, and transferability [53], along with reflexivity [54].

## 3. Results

There was a breadth of definitions given for peer support across universities. When describing a 'peer,' most focused on shared experiences or identities, but some centred on the difference (i.e., year of study) to define 'peers.' Peer support was widely defined as a student-led, collaborative space focused on support.

### 3.1. What Is a Peer?

Participants defined a peer as a fellow student at their university, with the shared experience of being a student forming the point of connection.

> ...it's...the journey of a student...it's...quite different from another same-age student who isn't at university, so it's having people that have been on the same journey...it's kind of 'been there, done that, got the T-shirt. Your T-shirt might be a different shape, size, colour, whatever to mine, but...we're still wearing the T-shirt... (Participant 14B)

This participant described being a student as a unique, shared experience. However, most participants did not explain why being a student was enough to define a peer.

Beyond 'being a student,' defining a peer focused on experiences or identities that brought peer facilitators together with the students accessing support. Participants drew in other characteristics that could be important, including course and shared identity or experience. For example, two universities provided peer support based on the shared identity of being part of the LGBTQIA+ community. Others discussed coordinating peer support for students living off campus or in university accommodation, international students, mature students, STEM students, student parents, working students, and study abroad students. Aside from defining peers based on being an international [55] or mature [56] student, these were novel shared experiences and identities not previously represented in the literature on university peer support for student mental health and wellbeing [26].

Some definitions of a peer, however, centred around differences. Many programmes defined a peer by year of study where students from different year groups were paired as peers. The students bringing different levels of experience was vital for this approach. New or lower-year students were paired with higher-year students to benefit from their experiential knowledge.

While some identities and experiences were novel to this study, defining a peer by course and year aligns with previous findings [26]. Many peer support programmes remain within subject disciplines and support the needs of new students transitioning into university [56]. Interestingly, approaches to defining a peer in the UK appear more narrowly focused than in international studies where peers have been defined by sharing experiences with mental illness [26]. Some UK studies have evaluated peer support for students with mental health difficulties where they were encouraged to attend a peer-led support group; however, the peer facilitators did not explicitly have the same lived experience so this was not an identity used to define a peer [16].

This study represented a considerable breadth of identities when defining a peer beyond being a student. A participant described this as a 'lived experience':

> ...it's just students using their lived experience to create a sense of community... it's... 'been there done that' kind of sharing process that I think is really, really valuable... (Participant 5)

Peers in university settings were, therefore, students who shared their lived experiences. While characteristics like academic department or identity were shared traits, the students' unique experiences or years of study varied when defining a peer. All were students, so the sharing process between them was valuable, no matter the additional identities or experiences. The breadth of definitions overall, both in the literature and this study, highlights the diverse types of students that peer support might help.

*3.2. What Is Peer Support?*

Participants discussed peer support as a student-led, collaborative space focused on supporting peers, according to the definitions of a peer established. Peer support was defined as being student-led. A sense of agency was highlighted as necessary, where 'students are able to work with each other on particular issues or topics that they want to cover' (Participant 1). Students leading the decision-making of how the programmes are run and the direction of sessions was a key theme in defining peer support. Other studies have also highlighted the student-led nature of peer support [56]. As peer support is student-led, defining a peer is an essential consideration. The definition of a peer can change how peer support is defined in practice. For example, peer support for LGBTQIA+ students may look different than for mature students.

In addition to being a space where peers could support each other and that was student-led, peer support was also defined as being collaborative. A participant described peer support as 'being kind of that space that you're with fellow students experiencing...similar... situations, and having that chance to think about how to support one another and support yourself' (Participant 8). Peers could talk about shared experiences and learn from each other, making it a particularly collaborative space. Peer support is defined as people supporting each other based on shared experiences [20] and students helping each other [26]. The emphasis on mutually supporting one another aligns with the definition outlined by participants in this study.

The definition of peer support generated through these interviews is broad, with consensus around an intervention that is student-led, collaborative, and about supporting peers. While inclusive, this open definition raises challenges for evaluating and comparing the efficacy of interventions. The exact details of how this form of support is implemented are essential. Universities should consider how they define a peer, as this specificity helps further describe peer support. In addition, different types of peer support are defined by their support structure. While using the broad definition, both the definition of a peer and the type of peer support should be made explicit when undertaking evaluation to make better comparisons. Types of peer support will be discussed next.

*3.3. Approaches: Types and Structure of Peer Support*

Five types of peer support were categorised: peer mentoring, peer-led support groups, peer learning, one-to-one peer support, and uncategorised. The researcher assigned each peer support programme a type based on its described structure. Although three types were recently outlined in the literature [23], no categorisations of all peer support types for student mental health and wellbeing exist. Consequently, staff referred to their programmes in myriad ways, which the researcher categorised according to descriptions of structure given. This process was sometimes challenging, as evidenced by the 'uncategorised' type. The supplementary material (Supplementary Table S2) outlines how each participant referred to their peer support programme and compares their terminology to the researcher's categorisation. The following outlines type definitions, with Table 2 giving a summary of key definitions and Table 3 outlining the frequency of each type by university and campus categories. Of the 31 peer support programmes, two were for postgraduate students only, seven were for postgraduate and undergraduates students, and the remaining programmes (*n* = 22) were for undergraduates.

**Table 2.** Summary of Definitions.

| Peer Support Term | Definition |
| --- | --- |
| Student Peer | A fellow university student who may share additional lived experiences or identities (i.e., living off campus, studying abroad, international, mature, LGBTQIA+, parent, being on same course, etc.) or who may be from a different year of study. |

**Table 2.** *Cont.*

| Peer Support Term | Definition |
|---|---|
| Peer Mentoring | Higher year or more experienced student peers supporting lower year or less experienced students. |
| Peer-Led Support Groups | A group of student peers gathered for mutual support. |
| Peer Learning | Convening student peers based on academic objectives. |
| One-to-one Peer Support | One-to-one approach for student peers in need of support. |
| Uncategorised | A peer support type described by participants that researcher was not able to categorise because of staff member presence. |

**Table 3.** Peer Support by University Category.

| | Ancient (*n* = 2) | Federal (*n* = 1) | International (*n* = 1) | Plate Glass (*n* = 5) | Post-92 (*n* = 3) | Russell Group (*n* = 2) | | 14 Universities |
|---|---|---|---|---|---|---|---|---|
| City-Based or Campus-Based | City (All) | City | City | Campus (All) | City (All) | City-Based (*n* = 1) | Campus-Based (*n* = 1) | City = 8 Campus = 6 |
| Number of Peer Mentoring | 1 | 0 | 1 | 7 | 1 | 2 | 0 | 12 |
| Number of Peer Learning | 1 | 0 | 0 | 3 | 0 | 1 | 1 | 6 |
| Number of Peer-Led Support Groups | 1 | 1 (attempted) | 1 | 2 | 2 (1 attempted) | 0 | 0 | 7 |
| Number of One-to-One Peer Support | 1 | 0 | 0 | 2 | 0 | 1 | 0 | 4 |
| Number of Uncategorised Peer Support | 0 | 0 | 0 | 1 | 1 | 0 | 0 | 2 |
| Total Number of Peer Support Programmes | 4 | 1 (attempted) | 2 | 15 | 4 (1 attempted) | 4 | 1 | 31 (2 attempted) |
| Russell Group (RG) | All RG | - | - | 2 RG | | | | |

### 3.3.1. Peer Mentoring

Peer mentoring was the most common type of peer support. Peer mentoring in a university context is defined as higher-year or more-experienced students supporting lower-year or less-experienced students [26]. Most programmes connected students with mentors based on their year of study, with many higher-year students being paired with lower-year, mostly incoming students. Three programmes brought students together based on additional shared experiences or identities (i.e., living off campus, studying abroad, and being part of the LGBTQIA+ community).

### 3.3.2. Peer-Led Support Groups

Peer-led support groups were the second-most common type. Peer-led support groups are a type of peer support that 'gathers groups of students for mutual support' [26]. The content discussed within the groups were either structured (i.e., a workbook is followed) or unstructured (i.e., the group decides the conversation that is not preset). Despite all being groups, three of the seven peer-led support groups operated differently. One was an unstructured group for PGR (Postgraduate Research) students with a conversation facilitated by PGR leaders (Participant 4). The other unstructured group was for students with specific experiences (LGBTQIA+, parents, etc.), but the discussion was open for all attendees to decide (Participant 10). The other event-focused one was for students from specific schools, with the direction decided by peer leaders (Participant 5).

The remaining four peer-led support groups were affiliated with Student Minds and were structured, set programmes (Participants 8, 9, 12, 14A). Student Minds trained all the participants to deliver training, supervise peer facilitators, and give ongoing support. Two participants could not start the programmes after attending the training. Their experience will be discussed in the 'challenges' section. The other two used the programme called Positive Minds, which is a structured peer-led support group for students who self-identified

as experiencing low mood or mild depression [57]. Students were given a questionnaire to decide if the group would suit them, but completing it was not required.

Overall, the group setting defines peer-led support groups. Nonetheless, many differences existed within this type of peer support.

### 3.3.3. Peer Learning

Peer-led support groups aim for mutual support to reduce isolation and improve mental health and wellbeing, but peer learning activities convene students based on academic objectives [26]. Although they often operate in group settings, this is not always true. The additional aims or benefits to mental health and wellbeing beyond learning outcomes were described by participants. Six peer learning programmes were discussed and referred to as Peer-Assisted Study Support (PASS), Peer-Assisted Learning (PAL), or Student Academic Mentors (SAMs). All peer learning had trained peer facilitators running the group sessions.

### 3.3.4. One-to-One Peer Support

One-to-one peer support was the least common type. This type was defined by its one-to-one structure and was not previously outlined in the literature [23]. Another defining feature of this type of peer support was that it was intended for students who self-identified as needing help. All were also affiliated with a student service team, such as counselling, wellbeing and welfare, or advice and support, in contrast to the other types of peer support, which were part of a diverse range of teams.

Two programmes (Participants 2 and 11) were developed on the 'Connect Peer Support' model developed at Oxford University through the University Counselling Service [58]. The two other one-to-one programmes (Participants 14A and 14B) mainly ran virtually. One was a Nightline support service, a student-run listening and information service open at night during term time in universities across the UK [59]. The volunteers are available via the phone, texts, instant messaging, and, for some universities, in person [59]. As volunteers are anonymous, they may not be a student at the same university or a student at all. These features make Nightline challenging to categorise as a specific type of peer support.

The other one-to-one programme was set up in response to the pandemic as a virtual offering where students could schedule a video call with a trained peer. The service filled the gap during COVID-19 of a previous peer support programme, which operated in halls of residence face-to-face. After moving back in-person, the team decided that it remained a helpful, more accessible addition to their service as they were reaching different students than their accommodation-based peer support programme:

> we are able to connect with students living off campus, and we found that's been brilliant, because we've had a lot of contact from students who are feeling isolated and just feeling a bit lonely, which maybe is more of a function of the last year (Participant 14B)

The structure of one-to-one peer support varied. Whilst three of the four programmes followed standardised structures and training developed by external partners, the third evolved based on the needs of the university during COVID-19.

### 3.3.5. Uncategorised Peer Support

Two types of peer support were not classified according to those outlined and are novel to this study [23]. Both could not be categorised because of issues with defining a 'peer.' Participants described peer support as a space for students only. Participant 7 ran what they classed as peer support, but the groups were called Reflective Practice Groups (RPGs) and had a staff member present in them. As the students were encouraged to support each other in the group, categorising this type as a peer-led support group was considered; however, the presence of a staff member changed the dynamic. One may argue that despite the staff members' presence, the space was still a student-led, collaborative space where students supported each other. Participant 14A coordinated a programme where students were trained to look after drunk students in a nightclub, but once again,

professional paramedics were on-hand if needed. Due to these discrepancies, it was not possible to categorise the programmes. Nonetheless, the participants did describe them both as types of peer support.

No prior study defined the various types of university peer support that aim to improve student mental health and wellbeing. Experts report the lack of formal definitions in university peer support programmes for student mental health as a barrier to the growth of this practice [30]. Whilst peer mentoring, peer-led support groups, and peer learning were found in the literature [26], one-to-one peer support and the uncategorised types were novel to this study. These findings build a fuller picture of the definitions and types of peer support in practice alongside the systematic review [26]. Researchers and practitioners should use the types outlined here to evaluate and report on effectiveness.

*3.4. Challenges and Lessons Learnt*

Challenges were discussed in response to a specific question, but the participants also volunteered them without being asked. The following themes emerged: engagement, resource and capacity, and evaluation. Whilst the challenges of peer support created barriers, participants had learnt lessons and explored solutions. These will be discussed in alignment with relevant literature and themes.

3.4.1. Engagement

'Engagement' was a theme that had multiple meanings. Whilst it was about students accessing peer support, it also referred to retention. For peer-led support groups, attendance was discussed as an issue affecting the nature of the group sessions, even after transitioning the sessions online in response to COVID-19:

> *. . .we had basically no attendance. . .we transitioned to doing peer support online, and we did actually have attendees. Generally, it was a single person attending. . .. the training was designed for group sessions in person, and we had to have some conversations about how do you do a one-to-one session online, and that presented its own kind of challenges that we had to work through (Participant 12)*

Attendance issues changed group sessions into one-to-one peer support for this participant, which they recognised required different training. Peer mentoring also experienced issues with engagement, especially retention:

> *. . .each mentor would have a few mentees, and it would be a group setting, and they met a lot in. . .the beginning of the year. . .they had a lot of questions, but (it) kind of just fizzled out across the first term. . .from what I gathered from the students. . . It was kind of something that happened over the first few weeks, but it wasn't something that they really sought support from across the year. (Participant 2B)*

Peer learning as a type of peer support, which was mostly for first-year undergraduate students, discussed challenges with engagement less. As outlined by Participant 5, '. . .peer learning is very well attended, and that is always like a really clear sign of success.' However, many did not run peer learning in the second semester because attendance was low. Some participants trialled moving the peer learning groups into one-to-one peer mentoring in the second term to maintain engagement. However, this did not work, demonstrating that retention beyond the first term remained challenging.

> *We've piloted doing some support in the second semester. . .we called them peer mentors. . . where we would buddy people up just to kind of have a bit of extra support, and we basically got zero take up from it. . .I think after they had the first semester with peer learning, they (students) just didn't see the need to reach out to the students and have like one to one support. (Participant 13)*

Another participant echoed this challenge to maintain engagement beyond the first university term both with mentors and mentees in peer mentoring: 'we tend to lose engagement from mentors. . ..around November, December time in the majority of cases. . .also,

some of the mentees either don't know that the mentoring scheme continues for a year or feel okay so they're not responding to messages' (Participant 3).

Most retention issues were cited in the first term of the first year of an undergraduate student's academic journey. In contrast, peer-led support groups for any PGR student (Participant 4) and peer mentoring for incoming PGR students (Participant 1) did not discuss issues with retention. Indeed, the PGR mentoring programme provided a specific number of sessions ($n = 6$) to avoid reliance on a mentor during the transition into a PhD (Participant 1), which may have set expectations of attendance for that time period. It is possible that retention issues are due to the nature of many peer support programmes focusing on the transition into university for first-year undergraduates. Some students may not feel they need peer support once they are settled.

For the uncategorised peer support type, the Reflective Practice Groups, which were timetabled into students' weekly schedules so that attendance was an expectation, in-session engagement was challenging.

> *Sometimes the students are just not chatty or talkative. And, that's nobody's fault…there's no particularly pressing topics, or people have just had a tough week, and they're feeling…a bit sort of burned out, and they're just not…up to talk in that session. (Participant 7)*

Engagement in the form of attendance appears to be the least challenging for peer learning. However, it was a constant theme for all other categories of peer support. In addition, engagement during sessions and retaining student engagement beyond the first term were consistently raised as a challenge among the participants.

Participants were considering new ways to tackle the challenge of engagement. Two themes were highlighted as potential solutions: implementing various peer support types and structures to reach a diverse student population and embedding peer support into the curriculum in partnership with academic staff.

Participants suggested that offering different types of peer support allowed for improved engagement in their programmes. One participant noted that their programmes worked well together, which students valued and wanted to be part of:

> *…I think we've created a really nice structure where different programmes and different things that we run feed into one another… we are offering resources to a narrow field of students who see the value and want to engage, but those students are engaging fully. They're invested. (Participant 12)*

In this way, the different types of peer support worked well together. Although they only benefitted a 'narrow field' of students, those who accessed them were wholly engaged. Another participant highlighted the benefit of having different types of support for engagement, namely one-to-one and groups:

> *… 'are you more comfortable just in a casual group setting?' or 'is there something really focused that you want to talk about, and you feel better talking to one person about that?' …the aim for both of those groups to exist… they satisfy different sorts of things, so that's why the peer mentoring schemes tend to be a lot more like identity based and experiential… it can be a lot more difficult for those students sometimes to want to come and talk to someone…whereas the peer support groups are a lot more casual… you're not committing to anything by turning up, you just …(are) hopefully getting something out of it, even if that means that you never come again and that's OK. (Participant 5)*

According to these participants, offering different types of peer support, structured in various ways, offers many points of contact. Different types, in turn, improve engagement as they attract a broader range of students.

Other participants reported that offering different peer support types and structures after the pandemic was important. Participant 1 had always run groups because they could reach more students: '…if you've got three students who are leaders who can still reach 20 students, whereas if you've got three students who work one-to-one, you can only reach

another nine.' They discussed that groups built community: 'having these group sessions are really fabulous because the students can talk to one another really easily' (Participant 1). Although this participant had run groups for engagement and community building, they had to shift during the pandemic:

> ...my colleague...put together a one-to-one scheme that would still enable us to continue even in a pandemic... we were a little bit worried that it wasn't going to be as effective and that we were not reaching as many students. But the learning for us out of that was that some students appreciated it more...were able to get more out of it, so once we had kind of developed both types, there was absolutely no reason for why we should then go back and say 'Oh no, we're just going to be doing group-based.' (Participant 1)

Although the pandemic forced a change, the university learned from it and changed the peer support offering. This participant also recognised that some forms of peer support, such as one-to-one, naturally have less engagement or different objectives. Therefore, universities should consider what objectives they are trying to achieve so they can choose relevant peer support types and structures accordingly.

Engagement was a common issue highlighted in practice [26,39] along with retention. In a quality assessment [60] of peer support studies measuring mental health and wellbeing, only 16 of 28 maintained engagement and met the criteria of less than 20% loss to follow-up after baseline, of which one had 2 participants [26]. Further examples are seen in the literature where only 34% of the original 65 participants attended all parts of a peer-led support group course [16].

Although the engagement challenge is seen most in practice, the literature suggests some barriers that affect utilisation. With 65% of UK students under 24 [57], most of the population are young people, where belonging to peer groups is especially important [61]. Disclosing to peers may be a barrier because of the saliency of identity and peer groups for adolescents [62]. Young people can feel disconnected from their social identity and mental illness stereotypes; stigma could be a significant barrier to help-seeking [63], telling peers [24], or accessing peer support. Other findings indicate that barriers to help-seeking include being unaware of services and skepticism about effectiveness [64], which may also contribute to low engagement.

Participants recommended implementing various peer support types and structures to reach a diverse range of students and increase engagement. The literature agrees that 'one size does not fit all' when it comes to university peer support and that programmes should be tailored to fit student needs [56]. This idea stems from the understanding that some factors present barriers to students being able to 'participate, integrate and feel like they belong' [65] (p. 5). Those working in partnership with students should develop an awareness and understanding of their students' challenges [65]. Local context is especially relevant. Those from backgrounds traditionally underrepresented in universities were most at risk of poorer wellbeing during the COVID-19 pandemic [66]. Risk factors included identifying as LGBTQIA+, self-declaring a disability, and having a previous diagnosis of a mental health condition [66]. A flexible approach is therefore advocated for in peer support. Engagement opportunities should be dynamic so that 'hard to reach' students can be supported, with their complex and heterogenous needs in mind [56]. More research is needed to understand how different types of peer support for a diverse student population can enhance engagement and impact.

Aside from creating different types of peer support, others highlighted the need for having peer support embedded into the curriculum. For the participant aiming to offer peer learning access to 80% of the undergraduate population, institutional commitment and embedding it into the curriculum were crucial steps:

> ... the institution is now going to commit...to demonstrate how they have built peer support into their programme...we expect it to be seen in their curriculum, so that's...a sort of massive step forward for us... getting buy-in and lecturers in faculty saying 'yes, we want this, we see the value'...students on their programme saying 'we want this, can



*you implement it?'... we're hopefully...not that many years now away from having full undergraduate coverage. And we're also aiming to expand the postgraduate coverage as well. (Participant 4)*

This participant demonstrated the need to embed peer learning into the departmental curriculum for more students to have access. With this, it was hoped that more students would engage. Beyond embedding peer support, their strategy was to incorporate new elements into these established peer learning programmes.

*... my view was you're better off focusing on something like PAL, which is embedded into the structure and bringing a wellbeing element into that rather than trying to set up yet another group that's labelled peer support. (Participant 4)*

This participant reported that incorporating wellbeing elements into existing, timetabled peer learning could improve engagement with peer support for university students' mental health and wellbeing. Whilst this suggestion may improve attendance, it was not clear how the peer support coordinator went about embedding their programme into departments. Another participant outlined how they embedded peer support into curriculums and academic departments:

*... one of the key things to success is to have academic staff who are really on the ball when it comes to the PAL programme, and that they want to make it a success...in the timetabling, that they allow for us to come to induction events, that we have a presence...in the school and that all helps. So the better the connections are with the school, the better the programmes tend to run. (Participant 1)*

This participant found that relationships and buy-in from the schools and academic staff were critical to embedding peer support in the curriculum. Others agreed:

*...if we've got a staff member who really sees the benefit of this and gets excited about it, you know, even if in principle...I think that it's probably gonna do better in that collect were we've got a really invested staff member. (Participant 6)*

Embedding peer support into the curriculum is a model used in peer learning. For example, Supplemental Instruction Peer Assisted Study Sessions (SI-PASS) is a form of peer learning that is embedded into the curriculum pedagogy because of its academic objectives [67]. Peer learning reported a few issues with engagement in this study. In a review of 28 studies, peer learning that measured mental health and wellbeing as an outcome was only represented in three studies, which all reduced anxiety [26]. Due to its academic focus, universities may overlook the opportunity that peer learning presents for students' mental health and wellbeing. In a new toolkit, which provides guidance outlining how the curriculum can support wellbeing and learning, no mention of peer learning was made [68]. However, peer mentoring was recommended to support student wellbeing in the curriculum [68]. One study of a midwifery programme found that students wanted opportunities to connect with their course peers [69]. Some suggest a pastoral and social focus for initial transitional peer support and an academic emphasis being added in the second term [70]. No matter the type or timing, peer support may offer the chance for students to connect, which shows promise for improving student mental health and wellbeing in the curriculum. While some champion the 'academic sphere' as the best place to encourage engagement and foster a sense of belonging [65], more research is needed.

Overall, participants recommended improving engagement in peer support for student mental health and wellbeing by implementing various types and structures of peer support and embedding peer support into the curriculum in partnership with academic staff. As all these recommendations require staff time and capacity to implement, it is essential to consider this challenge next.

### 3.4.2. Resource and Capacity

Whilst peer support is student-led, staff time and capacity are required. Staff responsibilities varied by the university, but they typically tended to train peer facilitators, provide

ongoing support, secure funding, develop structures for students to thrive in their roles, promote peer support, collaborate with colleagues across the university, and evaluate. For example, students across all institutions received specific training, or in one instance (Participant 7), a handbook for their roles, which relied on staff creating or delivering it. Six participants also provided supervision or debriefing for peer facilitators. These same participants, along with three others, discussed their approach to mitigating risk and safeguarding students, which mostly entailed training peer facilitators to signpost to professional services, have boundaries for their own wellbeing, and know which staff members to alert if they are concerned about a fellow student. Staff providing this ongoing support was recognised as an area that might be challenging. Due to this, four participants discussed the supervision they received as staff. In this way, staff provided both practical and emotional support to peer facilitators through supervision and training so they could safely and effectively offer peer support.

With this, the student-led nature of peer support can be challenging for staff. Working with students, especially as volunteers, requires significant flexibility in adapting to students' changing availability and commitment levels, which participants highlighted. Staff provide continuity in running a student-led service that will inevitably have a revolving door of students delivering it. The time of staff to provide this support is critical to the success of programmes.

Having sufficient resources to increase staff capacity was a key theme of the challenges. The barrier inhibited two participants from getting their programmes started. In addition to blocking the launch of peer support, it was also a significant challenge for growing established programmes to reach more students.

The two participants who tried to launch the Student Minds peer-led support groups and attended the training indicated they could not give the time needed to start it. One explained that they knew more resource was needed in the form of at least a part-time staff member, but they did not have the time to write a business case for a role after attending the training. They explained:

> *. . .2019 turned out to be a really busy year. . .really there wasn't any opportunity. . .to put enough welly into persuading the executive team of the university that they needed to resource a peer support programme. (Participant 9)*

The other participant based in a counselling service had support from their manager but still lacked time to launch the peer support programme alongside being the only CBT (cognitive behaviour therapy) therapist with a growing wait list.

> *although. . .my manager was brilliant, and she was like, 'Take the time you need out to do it.' It was actually, in reality, hard to make that happen. (Participant 8)*

Both participants could not launch their programmes. More staff help was needed. Another participant who had a team of two staff members to support peer learning, peer mentoring, and a peer-led support group admitted that they needed more staff.

> *. . .to get to that full 100% coverage, it does need more people, but it's not just about that. It's like you can have a PAL scheme, and it operates, and it's there, but if very few students. . .are engaging with it, that needs time and resource to work on that programme to try to. . .rectify that. . .we're missing that because of a lack of resource at the moment. (Participant 4)*

This participant highlighted the need for staff to grow programmes across a university and improve engagement with them. This issue was shared with another participant who managed a central peer support team based in the student union for peer learning and peer mentoring across the university.

> *There have been conversations about new and different peer support models. We're not in a position with sort of staff resource to take on anything that's completely new because. . .of workload that is required to set that up. . . evaluate it and pilot it without having anything to kind of start with. (Participant 3)*

The lack of resources to increase staff capacity inhibited peer support programmes from being launched and stopped growth: 'I think in terms of continued progress; it's always going to be manpower' (Participant 1).

'Resource and capacity' was a theme focused on the staff or 'manpower' to support these programmes to run effectively. This theme demonstrates the critical role of university staff in coordinating and delivering university peer support. Whilst peer support is student-led, the participants demonstrated that a great deal of staff time and capacity is required to support students in leading this work.

Securing enough resources and capacity for staff to support the implementation, development, and growth of peer support was a challenge discussed more than potential solutions. Many needed funding from their institution to hire more staff. In the case of one programme that could not launch, a suggestion was offered. When they completed training, the participant wanted to create a business case to request more capacity and resources to launch the peer-led support. However, they could not find relevant evidence to support their business case. They wanted to know that the peer-led support groups filled a gap they could demonstrate to their institution:

> . . .they (the institution) would want to know. . .that they were being used by students who need support but may not access other types of support. That this was doing something different from what. . .everything else we're doing. (Participant 9)

In addition to accessing evidence for peer support, the participant also needed a better understanding of how much staff time it took to run a programme. When considering what they might have done differently before attending the training and attempting to setup a peer-led support group, they described:

> . . .I would have had an understanding before the training. . . what is the time commitment that most people have found is necessary after doing the training to put this into practice? You know, in hard FTE terms. . .is it one person's job for six months or. . .we would have been able to discuss with him (the manager). . . 'realistically, we're gonna have to take one of us out of action for two days a week for the next three months to make this a reality,' or whatever it would be, and have had that conversation before we did the training. (Participant 9)

To gain more resources and capacity, especially for new programmes trying to launch, it may be worthwhile for researchers to work with practitioners or organisations like Student Minds to develop a business case for different types of peer support, outlining the available evidence and what staff support is needed.

The findings of this study provide a helpful start to understanding the '*hard FTE terms*'. Within the 14 universities, 33.3 full-time equivalent (FTE) staff were across 31 programmes, which equated to roughly one FTE role per programme, with 2.2 types of peer support on average at each university. However, many staff members did more than coordinate peer support (i.e., teaching, counselling, etc.). More than half of the staff (17.3 FTE) focused on peer support only in their roles; all but one were part of central teams that offered peer support across the university. Across the 12 programmes where staff only coordinated peer support, an average of 1.4 FTE staff supported each programme. Considering these staff only did peer support, more time is required than the average 1 FTE role observed across all 31 programmes. Since the staff solely focused on peer support still discussed the need for more capacity, universities should plan for each programme to require 1.5 FTE time dedicated to peer support. This need is especially relevant for universities wanting to provide peer support centrally to all students, as all but one of the dedicated roles did this. More evidence on peer support must be provided with these findings to complete the recommended business case.

Another possible solution was to work together better. When one participant who was unsuccessful at setting up the peer-led support group was asked what they would do differently, they wished they had gotten others on board across the institution to help.

*...I would have to be...clear 'like...this is your responsibility for this part of it. This is the responsibility that I hold. This is what you're doing. This is what I'm doing.' And, kind of try and make it much more of a joint venture... the...relationship with the SU would need to be a lot tighter from the offset...I just think like they're key, really, to getting students involved... (Participant 8)*

In this case, the participants did not say they needed more staff. They wished they had worked with others better to help alleviate the resource and capacity issues. They felt they could do this next time if more explicit expectations were set. Overall, the challenge of resources and capacity could potentially be improved according to the participants through access to a business case for peer support and collaboration across the university with clear expectations.

Peer support has historically been viewed as a cost-effective intervention. For example, in the case of peer education, the economic viability is attributed to the low cost of training educators compared to the potential size of the population that may be reached [71]. Similarly, peer education tends to rely on volunteers [72], making it less expensive than professional interventions. Other forms of peer support are similar. Only two universities in this study discussed paying their peer facilitators. Whilst this may be the case, others stress that peer support is not a 'cheap' alternative to professional services [34]. Programmes' sustainability, effectiveness, and safety rely heavily on individuals coordinating and supporting them [34]. The findings of this study support this idea. They especially indicated that universities financing peer support overlook the resources and capacity required from staff.

Further investment in peer support is required [34]. While peer support might be a relatively low-cost intervention, it cannot be run without cost. Currently, peer support interventions seem to be coordinated on such a tight budget that there is insufficient staff to improve quality and growth. Staff do not have the time to develop training to ensure high-quality support or the capacity to embed peer support into the university support ecology so that interventions can grow sustainably. Consequently, students lack awareness of peer support or its potential benefits. Without resourcing, peer support interventions might be doomed to fail, with poor engagement and retention perpetuated. It also has an impact on evaluation, which will be discussed next.

### 3.4.3. Evaluation

The final theme was evaluation– knowing what to measure and how to undertake it. One objective of peer support agreed upon by participants was engagement. As Participant 12 described: 'I hope that it is a group that is attended by at least maybe a handful of students...' It was logical then that most participants measured engagement as a key objective. While many kept track of engagement numbers for those accessing peer support and for the peer facilitators, there were questions raised if it was a good measure of success:

*...is success a big number? ... is success supporting a student to succeed and develop?... It's quite hard to...see...the true impact 'cause we don't see it every step of the student journey... if you say to students... 'why don't you contact X service' to then find out if they go to X service...And then find out what they do from there...actually they don't need to go to a service. They just needed to have a chat. So I think success is really hard to define. (Participant 14B)*

The participant highlights the difficulty of defining success and measuring it. They raised questions about whether success might be connected to students' development or reaching out for professional help. They acknowledged, though, that these are complex things to measure. The participant particularly highlighted this challenge when evaluating if peer support helped a student access a service because of the supportive chat or if the conversation itself was all they needed. The challenges of defining and evaluating success were shared.

*So, at the moment, we're measuring how many students we're reaching, but perhaps not how we're reaching them...and the impact of something like PAL is often ephemeral, things that cannot be measured as easily. And I think that's important. (Participant 1)*

This participant also recognised the evaluation challenges in defining and measuring impact. Especially in the case of PAL, they recognised that the impact of peer support may be short-lived, making its evaluation even more difficult. In the case of peer mentoring and peer-led support groups, similar challenges were raised:

*...it's a nightmare to report on this sort of stuff 'cause everything is so nuanced...that is the way peer support should be because it's about individual...or group experience, but it makes it really hard to find a narrative of what works...If they come to a session and they're like 'OK, either I've got what I needed' or identify that this isn't really right for me ... it does...mean that it's very hard for us to measure a benchmark of... how well received it was...we would be focusing a lot on how the peer mentors themselves felt about the process or whether they felt they were supported in it. (Participant 5)*

The participant further defined the challenges of evaluating peer support, especially if a student decided they got what they needed from it before it was finished. They also discussed the importance of evaluating the peer facilitators' experience in finding the 'narrative of what works.' Participants wanted to measure impact beyond engagement numbers but found it challenging.

Another objective of peer support highlighted by a majority of participants was building a community where students felt supported by their peers. One where they felt a sense of connection and belonging. When asked what success looked like for peer support, Participant 11 stated: '...we're about the connection.' Another participant shared: 'I hope that (peer support)...is something that they (the students) can benefit from and find a sense of community' (Participant 12). While creating a sense of community belonging as a primary objective will have a secondary outcome on mental health [73], no participants highlighted student mental health and wellbeing as a primary objective.

Without wellbeing or mental health being an explicit primary outcome, it was not surprising that few measured it. Although all participants self-identified their peer support programme as supporting student mental health and wellbeing during recruitment, most participants found measuring student mental health and wellbeing a challenge. When asked if they evaluated mental health and wellbeing, one peer learning programme participant shared:

*We don't have...anything around that... I think having a question in there without probing too deeply into (a) student, that would be quite useful 'cause...students find it hard, I think, to admit sometimes that they need help or that they've been struggling. So it's kind of how to delicately phrase that question, really. (Participant 13)*

They went on to explain that anecdotal feedback received during recruitment was where they heard most about peer learning reducing loneliness, which was also not measured. Although they did not measure mental health and wellbeing, they wanted to. However, participants were unsure how to measure the impact of peer support on mental health and wellbeing without pressing too much into an uncomfortable topic for students.

There were two challenges with evaluation: uncertainty of what to measure and a lack of evaluation for mental health or wellbeing outcomes. Lessons learnt included using a mixed-method evaluation approach and trialling specific measures. One participant who raised how difficult it was to measure peer support beyond engagement numbers suggested that more student narratives might be helpful. They explained: 'In some ways, more qualitative data would be quite useful' (Participant 1). Multiple participants discussed the qualitative feedback that they gathered to demonstrate impact. From free text boxes asking what the student found helpful (Participant 7) to what they enjoyed about peer support (Participant 3), participants used the student perspective to get a sense of effectiveness. When asked what was going well, one participant discussed their feedback:

> . . .the anecdotal stories of 'my peer mentor really helped me with this' or 'I couldn't have got through this year without my peer mentor; they've helped me so much'. That is really great to see, and there is a lot of success. . .it's just a shame that we. . .haven't been able to have the wider evaluation of that yet. (Participant 3)

Gathering student stories was the most common way to understand impact; however, it was not formally evaluated. Another informal measure of success was the experience of peer facilitators.

> . . . success for us. . . because we do invest so much in the leaders, is how the leaders feel. . .not so much the attendees that they're reaching. . .a measure of success is like how valued the leaders have felt, how much they engage in our reward and recognition programmes. When they do the reflections for us at the end of the year, how did they feel about the role? (Participant 5)

According to this participant, the peer facilitators' feedback was critical for understanding their programme's success. Aside from gathering anecdotal feedback, participants were still learning how to approach the evaluation challenge. However, they agreed that different evaluation forms were required, which included conversations and multiple perspectives to gain a broader view of success. A holistic approach to evaluation was demonstrated by Participant 4:

> . . .debrief programmes are part of the evaluation design . . . we have the questionnaires for both the leaders and the students, and. . . we have our own staff observations from our interactions with them. But also we. . .go in and do observations and then we have. . .the academics, we ask for their feedback as well, and we triangulate and pull all of those together. . .that's how we evaluate.

Participants explored a range of perspectives, which may provide an approach to tackling the challenge of what should be measured.

Only a few participants discussed the primary or secondary mental health and wellbeing outcome measures they were using. Both were piloting these questions at the time of the interview, demonstrating the lack of reporting in this area. Participant 14B had recently added belonging questions in their evaluation, which they only added because their Strategic Planning Department wanted the information. The participant highlighted that they 'never thought to ask around that, but it's quite interesting' (Participant 14B). Although improving belonging as a primary outcome affects mental health secondarily [73], only one participant was directly measuring student mental health or wellbeing as a primary objective. Participant 4 was using the Warwick-Edinburgh Mental Wellbeing Scale, which they were not 'a huge fan of that as a. . .meaningful measure of the peer support element.' However, for this type of peer-led support group pilot, 'the evaluation did conclude that those groups were very well-received and students very much valued having the PGR Connect groups' (Participant 4). Most participants felt that the programmes were making a positive difference from feedback, but they were unsure how to approach this from an evaluation perspective. This feeling is captured by the participant's response when asked if they evaluated the success of their peer learning programme, which they outlined as making friends, reducing isolation, and preventing issues with mental health.

> . . .something that comes up in our. . .application process where we ask that question: 'Why have you applied?'—particularly from some of our international students, we get quite detailed answers where they say 'I came to a country I'd never been to before. Felt really lonely, didn't know what to do, and my peer leaders. . .lifted me up and gave me that safe space where I felt welcome.' So I think that's the only thing I could reference to. . . (Participant 13)

Aside from questions about belonging and a pilot of the Warwick-Edinburgh Mental Wellbeing scale, participants discussed only a few mental health and wellbeing measures, either as primary or secondary outcomes. Despite open qualitative approaches to evaluation, participants were still working to grapple with the evaluation challenge. One participant

working to reduce stigma and increase mental health literacy through their peer support programme was considering this challenge when a student shared their perspective.

> '. . .you have to think about it as planting a seed. You might not see the results for a while to come, but. . . that doesn't mean that you should stop doing it.' So with pretty much everything that we do, I try to take that mentality that it's a seed. And that we're trying to decrease stigma and increase literacy with every single person that kind of passes through our door. . . (Participant 12)

The power of peer support from a student peer perspective was that it sowed a seed that might not be possible to see the benefits immediately. For this participant, the perspective eased their worries about peer support having an impact, and helped them face the complexities of evaluating it.

The challenge of evaluation made demonstrating that peer support was achieving what it set out to do very difficult. In the case of peer education, the evaluation issue is well-documented, with the need for a model of peer education evaluation for efficacy identified 20 years ago [71]. Peer-based programmes have not had an extensive evaluation, so the evidence for effectiveness is lacking [34,74–76]. The proposed mixed methods approach is also recommended in peer education [76], but the best measures to use in the evaluation were unknown to the participants. This finding builds on a recent review that collated the mental health and wellbeing measures used in peer support interventions. In it [26], the following outcomes and measures were used most: stress using the Perceived Stress Scale (PSS) [77], anxiety with the State-Trait Anxiety Inventory (STAI) [78], and loneliness using only one measure, the revised University of California—Los Angeles Loneliness Scale (UCLA) [79]. Other outcomes were depression and wellbeing, but all measures were used only once [26]. For depression, a few examples of measures were the Depression, Anxiety, and Stress Scale (DASS-21) [80], and the Primary Health Questionnaire (PHQ-9) [81]. A recent report recommended measures for wellbeing in the university student populations [82]. Whilst the authors could not recommend one single measure, they did note that the Warwick-Edinburgh Mental Wellbeing Scale (WEMWBS) [83,84] and the General Population-Clinical Outcome Routine Evaluation (GP-CORE) [85] demonstrate value [82]. They both have validation in the UK student population and offer comparability with the extant literature [82]. Universities can use this shortlist of measures to decide how they will evaluate outcomes.

The themes found here of engagement, resource and capacity, and evaluation were vital because they hindered peer support's development, growth, and success. Low engagement and lack of evaluation frameworks left staff coordinating these programmes in a difficult position. While they wanted to tackle these challenges, the lack of resources and capacity meant they had no time to do so. Without a way to demonstrate effectiveness, asking for the investment required to improve these programmes was even more challenging.

## 4. Conclusions

Peer support may be an innovative intervention to support students. However, the sector has lacked consensus on definitions and types of university peer support for student mental health and wellbeing to harness the potential of this approach. Little research has captured the experiences of staff who coordinate these programmes and have the practical expertise to offer. The study aimed to fill this gap in understanding through semi-structured interviews with 16 staff representing 14 UK universities.

Peer support was found to be a student-led, collaborative space focused on supporting fellow students. While this definition is broad, what it looks like in practice relies on further defining a 'peer' and clarifying the type of peer support being implemented. A 'peer' in university settings is defined primarily as a fellow student who may have additional experiences that unite them, such as course, year of study, or identity (i.e., international, LGBTQIA+, mature, etc.). In addition to the known types of peer support, peer-led support groups, peer mentoring, and peer learning, two additional types were found in practice: one-to-one peer support and uncategorised. These improved definitions and clarity around

university peer support can help the sector report on this intervention with more specificity and a shared language. With this guidance, a better understanding can be established through further research investigating which types of peer support are most effective for student mental health and wellbeing and for whom.

This study captured the unique perspective of staff who coordinated peer support, often behind the scenes. The challenges of engagement, resource and capacity, and evaluation were uncovered through conversations with staff representing a diverse range of universities. Staff found that offering different types of peer support and embedding it into the curriculum helped improve engagement. A mixed methods approach to evaluation and validated measures of mental health or wellbeing were also outlined. Above all, though, an investment in staff was required so they have the capacity to tackle these challenges. The study's findings provide sector guidance for further developing peer support as part of a whole university approach to student mental health and wellbeing, especially after the pandemic when their needs require urgent attention.

**Supplementary Materials:** The following supporting information can be downloaded at: https://www.mdpi.com/article/10.3390/educsci13090962/s1, S1: Topic Guide for Interviews; Table S1: Codebook; Table S2: Types and Categorisations of Peer Support.

**Author Contributions:** Conceptualisation, J.P.-H.; methodology, J.P.-H.; validation, J.P.-H., N.B., J.F., C.H. and J.O.; formal analysis, J.P.-H.; investigation, J.P.-H.; data curation, J.P.-H.; writing-original draft preparation, J.P.-H.; writing-review and editing, J.P.-H., N.B., J.F. and J.O.; visualisation, J.P.-H.; supervision, N.B., J.F. and J.O.; project administration, J.P.-H.; funding acquisition, N.B. and J.F. All authors have read and agreed to the published version of the manuscript.

**Funding:** This research was funded by the Economic and Social Research Council grant numbers ES/P000703/1 and ES/S00324X/1.

**Institutional Review Board Statement:** The study was conducted in accordance with the Declaration of Helsinki, and approved by the Research Ethics Office of King's College London (confirmation number MRSP-20/21-23031 approved on 29 April 2021).

**Informed Consent Statement:** Informed consent was obtained from all subjects involved in the study and for the findings to be published.

**Data Availability Statement:** Data are available from the corresponding author upon reasonable request.

**Conflicts of Interest:** The authors declare no conflict of interest.

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
