# Peer review of "Staff Perspectives: Defining the Types, Challenges and Lessons Learnt of University Peer Support for Student Mental Health and Wellbeing"

_education, doi:10.3390/educsci13090962_

Round 1

Reviewer 1 Report

Thank you for the opportunity to review this paper. Scholarship in the area of peer support is necessary. The following suggestions are made to help further strengthen the paper:

I feel that a figure depicting the findings would help to orient the reader in what is, by the nature of the research, text heavy. I found myself having to go back and forth a bit with the sub-headings to keep myself on track.

Abstract lines 13-15 - I think it would help clarity if it is made clearer in abstract that it is peer support for student health and wellbeing.

It would be helpful to clarify if both UG and Postgrad students are the focus of the work.

I am aware of some frameworks (e.g Brown, Duffy, Hill below) that situate peer interventions within overall mental health supports. Perhaps the authors might consider discussing their work with reference to such frameworks, or mention in the introduction?

Student mental health: some answers and more questions - PubMed (nih.gov)

Mental health care for university students: a way forward? - The Lancet Psychiatry

Student mental health and well-being: Overview and Future Directions - PubMed (nih.gov)

Line 182 - 184 Did all analysis occur after all interviews were completed?

Line 260 decision making in relation to what?

Line 321-322 I am unclear about this, group structure and / or group content?

In section 3.4.1 timing of supports is discussed eg line 447 engagement beyond the first term. Is this the first term of first year? Were other transitions mentioned eg timing of support for Y1 to Y2, finishing UG study, commencing PGR study?

Line 436 - what about losing engagement from mentees? Mentor engagement is quite different?

Section 3.4.3 Can the authors discuss student mental health and wellbeing as a primary or secondary outcome eg. peer interventions with the primary objective of reducing loneliness will likely have a secondary outcome on wellbeing. or academic interventions will likely reduce anxiety etc etc.

Were all participants non-academic members of staff? Line 147-148

In places e.g. line 350 "halls" suggest writing halls of residence as halls may not be a term used outside the UK

Line 372 "in a club" is this a nightclub?

Section 3.4.2 - did supervision of peer leaders (staff or student) come up at all? Line 69 points to need to investigate training and support for peer supporters. Also did managing risk come up at all. Can peer support ever be damaging? There appears to be an assumption that all peer support is a good thing.

Reference list does not go past 57 but in-text citations are greater - up to 82.

Add in-text citation for the "recent report" line 885

Table 2: I would find the table easier to understand if the rows were the columns and vice versa, but maybe this is just me!

Some sentences were difficult to grasp.

Line 61/62

Line 101 - "for" seems incorrect word here?

Line 238 - 241

Line 288 ? should read referred

Line 573

Line 792-793

Line 891-892

Line 895-896

Line 904 suggest "this" in place of "the"

Reviewer 2 Report

Well articulated and the structure of the paper is sound. The topic is timely. I appreciate the detail provided in explaining the qualitative method. 

However, I believe the emphasis should not be on the definition of "peers" as it is clear that in a university setting a peer refers to a fellow student. A more worthy exploration might be whether peer support is effective or not compared with other types of support offered by the university, for example, mental health services. Among the different types of peer support, which type is more effective than the others? The answer to this question would have more applied significance than definitions. 

Reviewer 3 Report

Dear Authors,

I really appreciate the opportunity to review the article titled "Staff Perspectives: Defining the types, challenges and lessons learnt of university peer support for student mental health and wellbeing," which has been submitted for consideration for publication in the journal Education Sciences. The current study has no flaws in ethics, trial design, methods, or statistics. The study appears to follow relevant guidelines and provides an original contribution to the existing scientific literature. There are no flaws in the data presented, and there are no misleading or false conclusions. No plagiarism was detected. 

I recommend this article for publication in Education Sciences in its current form.

Sincerely,

Reviewer

Author Response

17 September 2023

Dear Reviewer 3,     

Thank you so much for taking the time to review our manuscript. Your feedback was very encouraging. We appreciated your recommendation for publication without corrections.

With thanks,